# OSS-Qual: Holistic Scale to Assess Customer Quality Perception When Buying Secondhand Products in Online Platforms

**Neus Vila-Brunet** [1,*] and **Josep Llach** [2]

1   Departament de Ciències Econòmiques i Socials, Universitat Internacional de Catalunya,
    08017 Barcelona, Spain
2   Department of Business Administration, Management and Product Design, Universitat de Girona,
    17003 Girona, Spain; Josep.llach@udg.edu
*   Correspondence: nvila@uic.es; Tel.: +65-1029-549-0034

**Abstract:** Online secondhand markets have been growing substantially over the last decade and are expected to grow further. In order to effectively promote the growth of online secondhand markets, this paper designs and validates a scale to assess customer perception of the service quality of secondhand products purchased via online platforms. Complementarily, the paper assesses how each of the different dimensions that configure the scale contributes to explaining the fulfillment of customers' expectations. The scale is defined by 23 items and is arranged in 5 dimensions from the literature on online commerce as well as on the sharing economy. A sample of 200 questionnaires is used for exploratory factor analysis. A second sample of 507 users is used for confirmatory factor analysis. The quality perceived by online customers of secondhand products depends on the quality of the interactions that they have with the website, with the vendor, and with the product. The dimension that contributes the most to customer fulfillment of expectations is product quality. Findings identify the items that contribute the most to quality perception and fulfillment of expectations, facilitating the development of more effective strategies for platform owners and vendors who want to attract and retain customers of secondhand products. Complementarily, these findings are useful to businesses and governments that want to promote a more sustainable economy by reducing consumption of new products and promoting reutilization of existing ones.

**Keywords:** secondhand; online; quality; fulfillment of expectations

## 1. Introduction

Secondhand shopping is defined as buying goods that have been previously owned by someone else [1]. Secondhand shopping was popular during the 18th and 19th centuries; however, during the 20th century, the growth of a wide array of new products at low prices put several flea markets at risk. Nonetheless, over the last 20 years, secondhand shopping has increased in popularity again [2]. Some of the latest data indicate that one-third of consumers have increased their secondhand purchases [3]. At the beginning of the 21st century, sales of secondhand products increased in brick-and-mortar retail stores, like thrift stores, and at flea markets. More recently, secondhand product sales have also gained market share in the online marketplace worldwide for budgeting reasons as well as for sustainability motives [4].

As of today, research on online secondhand markets is at its early stages; most of the literature is descriptive and at the explanatory phase [5]. Given the growth in sales of the online secondhand markets, there is a need to further identify the factors that enhance the quality perception of online secondhand shoppers [6]. Current quality scales do not fully capture the specificities of the online

secondhand market for the following three reasons: (a) most quality scales assess the quality of the service offered and do not include the dimension of tangibles or the dimension of the actual product [7,8], (b) the few quality scales that include the tangible or product dimension do not consider the extra inherent risk of secondhand products in their constructs [9], and (c) there are few quality scales that distinguish the impact of having two intermediaries during the transaction (platform and vendor) as occurs in most secondhand markets [9,10]. Therefore, there is a need to develop a quality scale for the online secondhand market with the aim of contributing to its success and growth.

A dimension that has not been investigated in secondhand shopping is that of identifying the perceived quality factors that contribute to customers' fulfillment of expectations. Fulfillment of expectations is understood as customers' perception that the outcome that they experienced from the beginning of the transaction until the end is the same as the one that they envisioned [11].

By expanding our understanding of what contributes to the quality level experienced by online secondhand shoppers, we can identify strategies that enhance customer attraction and retention. These strategies can be useful for existing online secondhand platforms as well as for online new goods e-retailers that want to expand their market share in the online secondhand market. Furthermore, the identification of the main factors that contribute to customers' fulfillment of expectations is key in order to rank and further tailor the strategies that can enhance specific business performance. Consequently, this paper has two main contributions. On one hand, it contributes to the secondhand online shopping literature by conceptualizing, developing, and validating a quality scale for online secondhand purchases. On the other hand, it identifies how each of the different dimensions that configure the scale contributes to explain the fulfillment of customers' expectations.

The concept of "quality" has evolved over time. Nowadays, there are two main approaches for the assessment of perceived quality that have been proven robust. On one hand, there is a general agreement that quality can be conceived based on the expectations generated about the experience and the perception that the individual has about the experience [12]. On the other hand, there is the performance-based approach presented by Cronin and Taylor [8].

Churchill [13] developed a robust methodology for defining and validating measurement scales. This methodology was conceived for marketing scales, and since then it has been adapted by many scholars in other areas of research [14–16]; results have been satisfactory. In the present paper, we apply Churchill's methodology, and we proceed by defining the constructs of the OSS-Qual (Online Second-hand Shoppers Quality) scale by defining the characteristics of the online secondhand market and reviewing the literature on quality scales related to online secondhand markets.

The website ease of use (WE) dimension is conceived based on the first scale that assesses the perceived quality of online services by customers. This scale is named ES-QUAL and was developed by Parasuraman and colleagues [12]. The scale identifies four dimensions, with one being efficiency. Efficiency considers the ease of using the web and processing the transaction on the part of the customer. This scale has been adapted by many scholars in different sectors, and it has proven to be robust in the assessment of customer-perceived quality [17–19]. Agarwal and Venkatesh [17]; McKinney, Yoon, and Zahedi [18]; and Cho [20] support that websites that are easy to navigate and that are appealing contribute to customers' perception of quality.

One example of the adaptation of ES-QUAL in the context of online platforms related to collaborative consumption services is the study of the determinants of satisfaction and repurchase intention conducted by Möhlmann [21]. Möhlmann's scale includes a dimension called service quality that assesses the quality of the services offered by a website, including how easy it is to process a transaction and how appealing the website design is. Möhlmann's results confirmed those from Parasuraman, Zeithaml, and Malhotra [12] showing that the ease of using the website is significant with respect to customers' perception of service quality, and they expand our understanding by confirming that the website's appealing design is significant in explaining perceived quality in collaborative consumption settings.

The dimension of platform services (PS) considers that the website offers all the main services needed by the customer [22], that the website has easy ordering and payment mechanisms [23], and that the platform is quick to respond to customers' questions [9,24]. Ert and colleagues [25] studied trust generation and reputation in the sharing economy. They focused on the role of personal photos in the context of Airbnb as signaling factors for trust generation. The study also assessed the role of ratings and reviews in trust generation and identified their significance. In a complementary study, Bapna and colleagues [26] also found customers' ratings and reviews to be significant for trust generation in online social networks. We include the platform service of offering the possibility to customers to leave public ratings and reviews on vendors as part of the platform service construct.

The dimension platform legal protection and trustworthiness (PL) has been studied by several scholars [9,10,27]. It includes aspects related to the perception of feeling safe during an online transaction and feeling comfortable about sharing personal information. A complementary aspect associated with the perception of trustworthiness is the perception that the platform provides reliable opinions of other customers. Cheng and colleagues created a scale that measures the quality in car-hailing services and identified the relevance of the following dimensions in consumers' perceived quality: structural assurance, platform responsiveness, information congruity, competence, and empathy [9]. Results identified platform structure assurance as significant in ensuring customer-perceived quality. Cheng and colleagues defined this dimension as the safeguards and legal structures that the platform provides. Complementarily, this team identified the dimension of information congruity as significant. This dimension is understood as the reliability of what is being explained and shown online with respect to what the customer actually gets. Marimon et al. [10] also confirmed the significance of this dimension in the CC-QUAL scale of collaborative consumption services.

The vendor quality (VE) dimension stems from the seminal paper of Parasuraman and colleagues in which they present the SERVQUAL scale. SERVQUAL measures the perceived service quality in offline settings and identifies five dimensions [7]. One dimension addresses the tangible aspects, and the other four dimensions relate to the characteristics of the vendor. The dimensions of responsiveness and assurance related to the vendor highlight that vendors have to respond at the expected times and be trustworthy and polite. SERVQUAL has been adapted and tested under several settings, and it has been proven robust in assessing customer-perceived quality. Ladhari presented a review of all the SERVQUAL findings [28]. This dimension considers that customers perceive that they can rely on the agreement with the seller [9,12], that they consider the vendor trustworthy [12], and that they perceive that the seller describes the product features correctly [12,21]. On another note, it also considers that the seller behaves professionally [12], responds at the right times [12], and has flexibility regarding when to meet up [9,12].

The dimension of product quality (PR) has not been studied in service quality scales since the output is a service and not a product. However, quality scales in sharing-economy services also include tangibles [10], or they include items on the scale associated with the quality of the vehicle used in the service or the quality of the apartment rented [23]. In this dimension, we consider that the pictures indicate that the product is in good condition, is in acceptable condition, can be used again several times, and offers a good price/quality deal.

The dimensions of the OSS-Qual scale identified from the literature are as follows:

- Website ease of use: the degree of user-friendliness offered by the website;
- Platform services: the added value that the services offered by the platform bring to the user;
- Platform legal protection and trustworthiness: the legal security provided and the trust generated by the platform;
- Vendor quality: the array and quality of services offered by the vendor;
- Product quality: the characteristics of the product that allow it to be used properly.

The complete table with the five dimensions and their items can be found in the Appendix A.

*Fulfillment of Expectations*

The assessment of services' perceived quality has gone hand in hand with the assessment of expectations since early on. Parasuraman and colleagues presented the SERVQUAL scale that used the perceived quality of the service together with the fulfillment of expectations as a means to estimate the quality of the service. Cronin and Taylor [8] reflected on the method proposed by Parasuraman and compared the SERVQUAL method with the performance-based method. Since then, the performance-based method of assessing perceived quality has been tested and proven in many sectors, and the concept of fulfillment of expectations has been conceived as a mediator between quality dimensions and consumer satisfaction. Marimon and his team assessed the role of fulfillment of expectations on higher education students and proved that the students' fulfillment of expectations plays a mediation role between the dimensions of quality and consumer satisfaction [29].

We understand the construct of fulfillment of expectations as a single-item indicator that assesses customers' fulfillment of expectations based on the experience that the customer has from the beginning of the transaction until the end [11]. In the case of the online secondhand market, the lapse of time between customers' generation of expectations and the actual transaction is short. Hence, in online secondhand shopping, customers' initial expectations remain the same during the transaction. Consequently, we assume that customers' perception of their fulfillment of expectations is accurate in our study.

The paper continues with the presentation of the study method and results, followed by a discussion of the implications for theory and for secondhand online management practice, and closes with the conclusions.

## 2. Materials and Methods

*Data Collection and Sample*

Two samples are considered in this study. The first one was used in the exploratory analysis of the dimensions, and the second sample was used for confirmatory purposes.

Respondents made a purchase of a secondhand product on an online platform within the preceding 12 months.

The first sample was restricted to 200 responses and was collected at the end of 2019. After the debugging exploratory analysis, the second sample of 507 responses was collected during February 2020. Both samples were composed of respondents from Spain, Portugal, Italy, and France in a proportional quantity.

The secondhand online market for the four European countries studied (France, Italy, Portugal, and Spain) is quite similar in terms of the population percentage buying in them. The four countries report that more than 30% of the population has made purchases in online secondhand markets [30–34]. Accordingly, the sample is considered homogeneous.

To ensure the validity of the responses, both process collections were conducted by a specialized external company. Table 1 presents the main descriptive statistics of both samples. There were no significant differences between the samples in terms of gender, age, and studies.

The questionnaire is structured in three main sections. The first section contains the personal information of the respondent. Before answering the second and third sections, respondents were advised that their responses had to refer to their last purchase of a secondhand product on an online platform.

The second and third sections include statements extracted from the literature. In these statements, respondents had to indicate their agreement/disagreement on a five-point Likert-type scale (1 meaning strongly disagree, 5 meaning strongly agree). The statements of the second section are about specific aspects of the purchase, and the statements of the third section are about the global perception of the respondent with the purchase. The statements that compose the dimensions of the scale are described in the Appendix A.

**Table 1.** Main descriptive statistics of the two independent samples.

| | EXP | CONF | | EXP | CONF | | EXP | CONF |
|---|---|---|---|---|---|---|---|---|
| **Gender** | | | **Age** | | | **Studies** | | |
| Male | 90 (45%) | 235 (46.4%) | 18–30 | 42 (21%) | 105 (20.7%) | No studies | 0 (0.0%) | 3 (0.59%) |
| Female | 110 (55%) | 272 (53.6%) | 31–40 | 51 (25.5%) | 120 (23.7%) | Primary | 9 (4.5%) | 14 (2.76%) |
| | | | 41–50 | 59 (29.5%) | 141 (27.8%) | Secondary | 55 (27.5%) | 139 (27.42%) |
| | | | >50 | 48 (24%) | 141 (27.8%) | Bachelor | 63 (31.5%) | 159 (31.36%) |
| | | | | | | Master | 17 (8.5%) | 57 (11.24%) |
| | | | | | | Postgraduate | 14 (7%) | 37 (7.30%) |
| | | | | | | Professional training | 42 (21%) | 98 (19.33%) |
| **TOTAL** | 200 (100%) | 507 (100%) | 200 (100%) | 507 (100%) | | 200 (100%) | 507 (100%) |

EXP: sample for exploratory purposes; CON: sample for confirmatory purposes.

## 3. Results

Results are organized in two subsections: firstly, the assessment of the OSS-Qual scale proposal is considered; then, the dimensions of the scale as antecedents of the fulfillment of customer's expectations are analyzed.

### 3.1. OSS-Qual Scale Proposal

Two steps were undertaken to assess the OSS-Qual scale proposal—an assessment of the measurement model and an assessment of the structural model.

Assessment of the Measurement Model: Validity and Consistency

Table 2 presents the exploratory factor analysis and the descriptive statistics of the items. Following the recommendations of Alonso-Almeida, Bernardo, Llach, and Marimon [35], four factors emerged with eigenvalues greater than 1 (Kaiser criterion). Only the items that (i) load minimum at 0.63 or more on a factor, (ii) do not load at more than 0.50 on two factors, and (iii) have an item-to-total correlation of more than 0.50 were retained.

In order to avoid common method bias, a Harman's single factor test was conducted following Malhotra, Kim, and Patil [36]. The variance extracted from the sample with the Harman's single factor test was under the recommended value of 50% which means no danger of common method bias.

The first factor, labeled "website", gathered (i) two items of platform services, (ii) two items of platform legal protection, and (iii) one item of website ease of use. The second factor, which consisted of all product quality items was labeled "product". The third factor gathered three out of seven vendor quality dimensions and was labeled "vendor". Finally, the last factor gathered (i) two items of platform services and (ii) one of platform legal protection. Since all the items were related to the importance of the evaluations of the service by the other customers, the factor was labeled "ratings and reviews".

The emergence of this dimension is in line with studies conducted in collaborative-economy settings that highlight the importance of ratings and reviews in order to reduce consumers' uncertainty and risk towards the vendor, as well as consumers' uncertainty and risk towards the item considered for purchase [37–39].

**Table 2.** Descriptive statistics and exploratory factor analysis (*n* = 200).

| | Descriptive Statistics | | Exploratory Factor Analysis | | | |
|---|---|---|---|---|---|---|
| **Item** | **Mean** | **St. Dev** | **1** | **2** | **3** | **4** |
| **PS3** | 3.805 | 0.884 | **0.747** | 0.243 | 0.238 | 0.056 |
| **PS1** | 3.770 | 0.872 | **0.720** | 0.368 | 0.177 | 0.096 |
| **PL2** | 3.655 | 0.975 | **0.712** | 0.140 | 0.272 | 0.226 |
| **WE3** | 3.660 | 0.847 | **0.699** | 0.187 | 0.309 | 0.144 |
| **PL1** | 3.600 | 0.919 | **0.678** | 0.203 | 0.312 | 0.086 |
| **PS2** | 3.800 | 0.951 | 0.618 | 0.241 | 0.055 | 0.416 |
| **WE2** | 3.950 | 0.788 | 0.617 | 0.461 | 0.048 | 0.239 |
| **WE1** | 4.140 | 0.790 | 0.604 | 0.403 | 0.163 | 0.086 |
| **PR3** | 3.960 | 0.788 | 0.279 | **0.817** | 0.208 | 0.124 |
| **PR4** | 3.920 | 0.779 | 0.279 | **0.795** | 0.207 | 0.168 |
| **PR2** | 3.960 | 0.788 | 0.306 | **0.788** | 0.224 | 0.178 |
| **PR5** | 3.940 | 0.806 | 0.286 | **0.754** | 0.245 | 0.235 |
| **PR1** | 3.920 | 0.753 | 0.240 | **0.742** | 0.251 | 0.187 |
| **VE4** | 3.730 | 0.788 | 0.223 | 0.225 | **0.735** | 0.219 |
| **VE6** | 3.650 | 0.807 | 0.184 | 0.423 | **0.661** | 0.188 |
| **VE5** | 3.410 | 0.988 | 0.334 | 0.005 | **0.636** | 0.439 |
| **VE7** | 3.793 | 0.801 | 0.129 | 0.483 | 0.600 | −0.079 |
| **VE3** | 3.735 | 0.836 | 0.302 | 0.366 | 0.604 | 0.149 |
| **VE2** | 3.570 | 0.836 | 0.466 | 0.077 | 0.587 | 0.384 |
| **VE1** | 3.690 | 0.829 | 0.366 | 0.320 | 0.553 | 0.326 |
| **PS5** | 3.690 | 1.024 | 0.137 | 0.205 | 0.155 | **0.826** |
| **PS4** | 3.625 | 0.921 | 0.045 | 0.246 | 0.387 | **0.726** |
| **PL3** | 3.590 | 0.925 | 0.418 | 0.149 | 0.152 | **0.659** |

To examine the one-dimensionality of these factors, four new independent exploratory factor analyses were performed, each with only the items that accomplished the criteria described above. Only one factor was extracted for each factor analysis (see Table 3).

Since all the loading values were greater than the cut-off level of 0.7 and had significant loads (t > 2.58), convergent validity was confirmed. In addition, Table 3 reports the internal consistency of the scales using Cronbach's alpha and composite reliability. Both values were higher than the cut-off levels of 0.7 [40,41]. Moreover, the Cronbach's alpha values did not improve when any of the items were removed from the scales for each dimension. Finally, the average variance extracted values were also over the cut-off level of 0.5, which is on the edge of that recommended by Fornell and Larcker [42].

**Table 3.** Loads of the five EFAs and statistics for their reliability analyses (*n* = 200).

|  | Website | | Product | | Vendor | | Ratings and Reviews | |
|---|---|---|---|---|---|---|---|---|
|  | PV3 | 0.803 | PR3 | 0.927 | VE4 | 0.857 | PV5 | 0.882 |
|  | PV1 | 0.824 | PR4 | 0.919 | VE5 | 0.812 | PV4 | 0.850 |
|  | PL2 | 0.798 | PR2 | 0.920 | VE6 | 0.811 | PL3 | 0.829 |
|  | WE3 | 0.804 | PR5 | 0.896 |  |  |  |  |
|  | PL1 | 0.820 |  |  |  |  |  |  |
| Cronbach's $\alpha$ | 0.868 | | 0.935 | | 0.760 | | 0.813 | |
| Range of Cronbach's $\alpha$ if one item is removed | 0.836–0.844 | | 0.910–0.925 | | 0.626–0.713 | | 0.695–0.780 | |
| CR | 0.905 | | 0.953 | | 0.866 | | 0.889 | |
| AVE | 0.655 | | 0.838 | | 0.683 | | 0.729–0.853 | |

CR: composite reliability; AVE: average variance extracted.

To demonstrate adequate discriminant validity of the dimensions, Table 4 presents the bivariate correlation between dimensions and the square root of average variance extracted (AVE) of each dimension in the diagonal. In all cases, the values of the square root of AVE were higher than the interfactor correlation values, and, therefore, discriminant validity was demonstrated [42]. The final scale for assessing customer perception of the service quality of secondhand products purchased via online platforms is presented in Appendix B.

**Table 4.** Discriminant validity.

|  | 1 | 2 | 3 | 4 |
|---|---|---|---|---|
| Website | *0.809* | | | |
| Product | 0.630 | *0.915* | | |
| Vendor | 0.618 | 0.566 | *0.826* | |
| Ratings and Reviews | 0.524 | 0.517 | 0.624 | *0.853* |

Square root of AVE in the diagonal; assessment of the structural model.

To set up the definitive scale, the second sample of 507 responses was used. Table 5 presents both the loading results of second-order confirmatory factor analysis (CFA) with structural equation modeling (SEM) using EQS 6.4 software [43] and the fit indices obtained.

**Table 5.** Confirmatory factor analysis (*n* = 507).

| Dimension | Item | Load | *t*-Value | $r^2$ | Sources |
|---|---|---|---|---|---|
| Website | WEB1 | 0.696 | - | 0.357 |  |
| | WEB2 | 0.790 | 11.61 | 0.477 |  |
| | WEB3 | 0.796 | 10.91 | 0.600 |  |
| | WEB4 | 0.750 | 10.86 | 0.447 |  |
| | WEB5 | 0.808 | 11.42 | 0.570 |  |
| Product | PROD1 | 0.835 | - | 0.582 | - |
| | PROD2 | 0.847 | 13.16 | 0.633 |  |
| | PROD3 | 0.855 | 11.96 | 0.638 |  |
| | PROD4 | 0.829 | 11.86 | 0.592 |  |

**Table 5.** *Cont.*

| Dimension | Item | Load | *t*-Value | $r^2$ | Sources |
|---|---|---|---|---|---|
| Vendor | VEND1 | 0.798 | - | 0.491 | |
| | VEND2 | 0.810 | 10.59 | 0.481 | |
| | VEND3 | 0.799 | 10.72 | 0.421 | |
| Ratings and Reviews | RR1 | 0.838 | - | 0.496 | |
| | RR2 | 0.852 | 10.59 | 0.586 | |
| | RR3 | 0.776 | 10.72 | 0.474 | |
| **Fit Indices** | **Value** | **Cut-off value** | | | |
| Satorra–Bentler scaled χ2/degree of freedom | 2.06 | <5 | | | Carmines and McIver (1981) Hair et al. (1998) Hu and Bentler (2009) Escrig-Tena and Bou-Llusar (2005) Alonso-Almeida et al. (2013) Alonso-Almeida and Llach (2019) |
| Bentler–Bonett non-normed fit index (BB-NFI) | 0.948 | >0.90 | | | |
| Comparative fit index (CFI) | 0.958 | >0.95 | | | |
| Root mean-square error of approximation (RMSEA) | 0.046 | <0.06 | | | |

The model was estimated using the robust maximum likelihood method from the asymptotic variance–covariance matrix. In all cases, the fit indices were over the cut-off values recommended by the literature.

According to the fit indices shown above, these measures of overall fitness reflect the explanatory power of the model. Therefore, it is demonstrated that there are four main dimensions that capture customer perceptions of service quality in the purchase of secondhand products on online platforms: (i) website, (ii) product, (iii) vendor, and (iv) ratings and reviews.

*3.2. Perceived Quality Dimensions of OSS-Qual Scale as Antecedents of the Fulfillment of Expectations*

A new model was proposed to understand which of the dimensions better explained the fulfillment of customers' expectations. Besides the correlations among the four dimensions, four direct relationships were established between these dimensions and the fulfillment of expectations.

The statistical process conducted to assess the model was identical to the validation process of the OSS-Qual scale. All the values of the fit indices were also over the cut-off levels and therefore satisfactory. The Satorra–Bentler scaled χ2/degrees of freedom was 1.943 (<5), BB-NNFI was 0.952 (>0.9), CFI was 0.962 (>0.95) and RMSEA was 0.043 (<0.06).

Figure 1 presents the standardized solution of the causal model. Standard coefficients of the direct relationships, covariances of the correlations, and t-values are included. The most important and unique antecedent of the fulfillment of expectations is the quality of the product; there is a direct and highly significant effect. None of the rest of the dimensions play significant role in explaining customers' fulfillment of expectations.

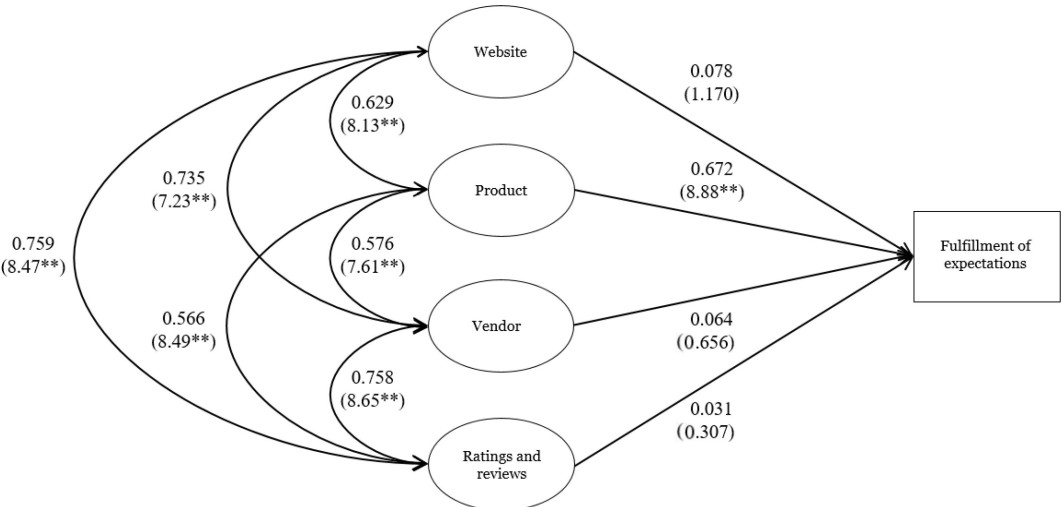

**Figure 1.** Standardized solutions of the causal model (*t*-statistics in parentheses; ** significant at *t*-level <0.01).

## 4. Discussion

### *4.1. Summary of Findings*

The paper presents the process of designing and validating an instrument to measure online secondhand consumers' perception of quality. The first step is to review and summarize the academic contributions to e-commerce service quality as well as the contributions to sharing-economy service quality. The next step is to identify the relevant quality dimensions for the online secondhand market. In order to identify the significant dimensions, we performed an exploratory analysis with a sample of 200 individuals. The significant dimensions were confirmed using SEM with a sample of 507 individuals. The final dimensions were (a) website, (b) ratings and reviews, (c) vendor, and (d) product. These dimensions indicate adequate reliability of the dimensions as well as adequate fit of the scale.

The paper assesses which of the OSS-Qual dimensions are significant in explaining online secondhand customers' fulfillment of expectations by using SEM. Results indicate that only the dimension of product quality is significant.

### *4.2. Implications and Future Research*

#### 4.2.1. Theoretical Implications

The theoretical contribution lies in the development of an instrument that measures online secondhand customers' perception of quality (OSS-Qual) and in the conceptualization of how customer-perceived quality contributes to customers' fulfillment of expectations.

One of the theoretical contributions that stems from the development of the quality instrument is the identification of its four relevant dimensions (website, ratings and reviews, vendor, and product). The structure of the OSS-Qual scale is similar to that of Parasuraman et al.'s ESQUAL scale [12], since it distinguishes the dimension of "website" that includes the ESQUAL constructs of efficiency, system availability, and privacy. Complementarily, OSS-Qual distinguishes the "vendor" dimension that assesses vendors' traits of responsiveness, assurance, and empathy, reflecting the SERVQUAL dimensions of Parasuraman et al. [12]. The two specific quality dimensions of OSS-Qual not considered in Parasuraman's previous studies are the "product" dimension and the "ratings and reviews" dimension. Parasuraman's scales focus on studying the delivery of services, which is why the product dimension was not included. However, the dimension of ratings and reviews is a new contribution made by the present study that is worth paying attention to in future research of service quality.

In the case of the online secondhand market, the results of the present study highlight that having access to vendors' ratings and reviews motivates customers to buy from them. The secondhand market has higher comparative levels of uncertainty and associated risk due to the fact that the product is secondhand. Consequently, having access to reliable ratings and reviews of the vendor, as well as to reliable opinions of other customers, is paramount in order to perceive the experiences as being of high quality.

The next relevant theoretical contribution stems from the identification of the OSS-Qual dimensions that contribute to customers' fulfillment of expectations. Out of the four dimensions of the scale, results indicate that only the product dimension is significant in explaining customers' fulfillment of expectations.

### 4.2.2. Practical Implications

OSS-Qual is useful for practitioners since it indicates the aspects that online secondhand customers value the most throughout the whole purchasing experience in order to consider it as high quality. Hence, on one hand, individuals, businesses, and governments who are already operating in the online secondhand market can use OSS-Qual to identify how they are performing with respect to each item. By doing so, current practitioners will be able to identify which of the items needs to be strengthened to enhance the perceived quality experience of its users, which items are performing regularly and need to be enhanced, and which items are performing excellently and require continuation. On the other hand, individuals, businesses, and governments that are considering expanding into the secondhand market can use OSS-Qual in their pilot projects to assess the quality of their websites and services provided before launching them. Furthermore, OSS-Qual items can guide new market entrants throughout the initial stage in order to fine-tune websites and the relationships with vendors and the products sold in order to make sure that they stay competitive and provide a high-quality experience to their users.

The finding that the only significant dimension that contributes to customers' fulfillment of expectations is product indicates that both platform owners and vendors should pay paramount attention to the following product factors: (a) pictures should clearly show the quality of the product, (b) the product has to offer a good price/quality deal, (c) the product has to be in acceptable condition, and (d) the product should be able to be used several times. Hence, from the platform perspective, if the aim is to enhance customers' perception of quality, rules have to specify that the only products allowed to be sold must comply with the above-mentioned features. From the vendor's perspective, the above-mentioned product features must be maintained in order to increase the possibilities of selling and obtain good ratings and reviews.

Last but not least, governments can promote the quality of the online secondhand market by taking into account the OSS-Qual dimensions and making the items that are being stored or underutilized in some facilities and warehouses accessible to the community. Complementarily, governments can assess the possibility of giving tax incentives to the platforms and vendors of secondhand products. By doing so, practices that promote a more sustainable economic system are encouraged and we move one step forward towards achieving the sustainable development goals.

### 4.2.3. Limitations and Future Research

OSS-Qual identifies two dimensions that have been seldom investigated in quality scales. These two dimensions are the "product" dimension and the "ratings and reviews" dimension. The results concerning fulfillment of expectations highlight that the only OSS-Qual dimension that contributes to customers' fulfillment of expectations is the "product quality" dimension. Hence, future research on online shoppers' quality experience should investigate the relevance and impact of these two dimensions on other sectors and industries.

The results presented come from a sample of consumers from France, Italy, Portugal, and Spain. The perception of quality in these four European countries is strongly connected to the quality standards achieved in these countries and to the culture of each of them. Hence, further research is needed in other regions that have different quality standards in order to identify whether different perceived quality standards generate a quality scale with the same dimensions or not and whether there is a substantial change in the significant items in them. Additionally, future research should investigate whether having a different quality perception baseline changes the results that identified product quality as the only significant quality construct that contributes to customers' fulfillment of expectations. In the same direction, further research has to assess whether culture plays a role in defining the dimensions and items of the quality scale and identifying the dimensions that contribute to customers' fulfillment of expectations.

Another important area of future research is to investigate whether there is a difference between transactions taking place in B2P and P2P markets in terms of quality scale constructs and items. The structure of the sample of the present study reports that 50% of secondhand online purchases are made from a business vendor (B2P) and 50% of the secondhand online purchases are made from a peer vendor (P2P). Future research needs to investigate whether there is a difference between the two market structures in the significant items of the OSS-Qual scale.

**Author Contributions:** Conceptualization, N.V.-B.; methodology, N.V.-B. and J.L.; software, J.L.; validation, J.L.; formal analysis, N.V.-B. and J.L.; investigation, N.V.-B. and J.L.; resources, N.V.-B. and J.L.; data curation, J.L.; writing—original draft preparation, N.V.-B. and J.L.; writing—review and editing, N.V.-B. and J.L.; visualization, N.V.-B. and J.L.; supervision, N.V.-B. and J.L.; project administration, N.V.-B. and J.L.; funding acquisition, J.L. All authors have read and agreed to the published version of the manuscript.

**Funding:** This article was written as part of a research project entitled "Improvement of quality in collaborative consumption companies: model, scale and loyalty (CC-QUAL)" (ref: RTI2018- 096279-B-I00), financed by the Spanish Ministry of Science, Innovation and Universities under the "Research Challenges" R&D Project aid program.

**Acknowledgments:** A previous version of the article was presented at the 4th International Conference on Quality Engineering and Management.

**Conflicts of Interest:** The authors declare no conflict of interest.

## Appendix A

**Table A1.** Initial online secondhand shopper quality scale dimensions and items.

| Dimension | Code | Statement/Item | Source (Adapted from) |
|---|---|---|---|
| Website ease of use | WE1 | The website is easy to use. | Parasuraman et al. (2005) |
| | WE2 | The website makes it easy to find what I need. | Parasuraman et al. (2005) |
| | WE3 | The website design is appealing to me. | Möhlmann (2015) |
| Platform services | PV1 | The website offers all the main services that I need. | Breidbach et al. (2014) |
| | PV2 | The website has easy ordering and payment mechanisms. | Clauss (2019) |
| | PV3 | The platform is quick to respond to my inquiries. | Cheng et al. (2018) |
| | PV4 | The seller ratings from other customers motivate me to buy from this seller. | Ert et al. (2016), Banpa et al. (2017) |
| | PV5 | The seller reviews from other customers motivate me to buy from this seller. | Ert et al. (2016), Banpa et al. (2017) |
| Platform legal protection and trustworthiness | PL1 | I feel comfortable about the privacy of my personal information. | Cheng et al. (2018), Clauss (2019) |
| | PL2 | I feel safe in my transactions. | Cheng et al. (2018), Clauss (2019) |
| | PL3 | The platform provides reliable opinions of other customers. | Marimon et al. (2019) |
| Vendor quality | VE1 | I rely on agreement with the seller. | Parasuraman et al. (1988), Cheng et al. (2018) |
| | VE2 | The seller is trustworthy. | Parasuraman et al. (1988) |
| | VE3 | The seller correctly describes the product features. | Parasuraman et al. (2005), Möhlman (2015), Cheng et al. (2018) |
| | VE4 | The seller responds at the right times. | Parasuraman et al. (1988) |
| | VE5 | The seller is professional. | Parasuraman et al. (1988) |
| | VE6 | The seller has flexibility regarding when to meet up. | Parasuraman et al. (1988), Cheng et al. (2018) |
| | VE7 | The quality of the face-to-face interaction with the seller is good. | Own proposal |
| Product quality | PR1 | Pictures indicate that the product is in good condition. | Own proposal |
| | PR2 | The product is in acceptable condition. | Clauss et al. (2018) |
| | PR3 | The product can be used again. | Own proposal |
| | PR4 | The product can be used again many times. | Own proposal |
| | PR5 | The product offers a good price/quality deal. | Own proposal |



## Appendix B

**Table A2.** OSS-Qual scale—final online secondhand shopper quality scale dimensions and items.

| Dimension | Code | Statement/Item | Source (Adaptation) |
|---|---|---|---|
| Website | WEB1 | The platform is quick to respond to my inquiries. | Cheng et al. (2018) |
| | WEB2 | The website offers all the main services that I need. | Breidbach et al. (2014) |
| | WEB3 | I feel safe in my transactions. | Cheng et al. (2018), Clauss (2019) |
| | WEB4 | The website design is appealing to me. | Möhlmann (2015) |
| | WEB5 | I feel comfortable about the privacy of my personal information. | Cheng et al. (2018), Clauss (2019) |
| Product quality | PROD1 | The product can be used again. | Own proposal |
| | PROD2 | The product can be used again many times. | Own proposal |
| | PROD3 | The product is in acceptable condition. | Clauss et al. (2018) |
| | PROD4 | The product offers a good price/quality deal. | Own proposal |
| Vendor quality | VEND1 | The seller responds at the right times. | Parasuraman et al. (1988) |
| | VEND2 | The seller is professional. | Parasuraman et al. (1988) |
| | VEND3 | The seller has flexibility regarding when to meet up. | Parasuraman et al. (1988), Cheng et al. (2018) |
| Ratings & Reviews | RR1 | The seller reviews from other customers motivate me to buy from this seller. | Ert et al. (2016), Banpa et al. (2017) |
| | RR2 | The seller ratings from other customers motivate me to buy from this seller. | Ert et al. (2016), Banpa et al. (2017) |
| | RR3 | The platform provides reliable opinions of other customers. | Marimon et al. (2019) |

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
