# Peer review of "OSS-Qual: Holistic Scale to Assess Customer Quality Perception When Buying Secondhand Products in Online Platforms"

_sustainability, doi:10.3390/su12219256_

Round 1

Reviewer 1 Report

This paper develops and validates a scale for assessing the perceived quality of online second hand websites. Overall, the paper is interesting and matches the subject coverage of this special issue. It represents, in my view, a valuable contribution to existing knowledge. I have though a number of suggestions for improvement:

  • How did you address common method variance bias?
  • The authors seem to confuse EFA with PCA, but they are not actually the same. You should have used Principal Axis Factoring. We also need information about the KMO test, Bartlett tests, variance explained.
  • In pag. 6, line 217, instead of table 4 should be table 3.
  • Please add an appendix with your final scale.

Reviewer 2 Report

  1. Although this study suggested that the quality of services about sharing economic services can be measured in five dimensions, there have no evidence about why these five aspects are important in sharing economic services.

Please provide theoretical background and evidence for measuring five aspects in a shared economy service.

  1. There has a question of whether the suggested quality model can be applied into all sharing economy platforms.

In car-sharing services such as Uber, the attitude of taxi drivers is important, and Airbnb needs to distinguish between trust in Host and trust in Airbnb. This study have to discuss the exact targets, and explain how these model can be transformed and used in other sharing economic platforms.

  1. In Figure 5, the items are not grouped into one consumers, but are grouped with other constructs. Although it is CFA, it is not understandable that many factors are grouped into different measurement items. (Ratings and reviews: PV and PL, Website:PV, PL, WE)

Round 2

Reviewer 1 Report

I checked the changes made by the authors. I now recommend acceptance for publication.

Reviewer 2 Report

This study reflected the reviewer's comments well.